# Predictors of Functional Improvement, Length of Stay, and Discharge Destination in the Context of an Assess and Restore Program in Hospitalized Older Adults

**DOI:** 10.3390/geriatrics7030050

**Published:** 2022-04-20

**Authors:** Beatrise Edelstein, Jillian Scandiffio

**Affiliations:** Humber River Hospital, Toronto, ON M3M 0A7, Canada; bedelstein@hrh.ca

**Keywords:** assess and restore, functional decline, length of stay, older adults, predictor

## Abstract

Assess and restore programs such as Humber’s Elderly Assess and Restore Team (HEART) provide short-term restorative care to prevent functional decline in hospitalized older adults. The aim of this retrospective observational study was to determine which HEART participant characteristics are predictive of functional improvement, decreased length of stay, return to home, and decreased readmission to hospital. Electronic health records were retrospectively examined to gather predictor data. Differences in functional status, excessive length of stay, discharge destination, and hospital readmissions were compared in 547 HEART patients and 547 matched eligible non-participants using ANOVAs, Mann–Whitney, and chi-square tests. The greatest functional improvements (percent Barthel change) were seen in those requiring a one-person assist (M = 39.56) and using a walker (M = 46.07). Difference in excessive length of stay between HEART and non-HEART participants was greatest in those who used a walker (Mdn = 3.80), required a one-person assist (Mdn = 2.00), had a high falls risk (Mdn = 1.80), and had either a lower urinary tract infection (Mdn = 2.25) or pneumonia (Mdn = 1.70). Predictor variables did not affect readmission to the hospital nor return to home. Predictive characteristics should be considered when enrolling patients to assess and restore programs for optimal clinical outcomes.

## 1. Introduction

Hospitalized, frail older adults are at risk for functional decline that could lead to iatrogenic complications, increased dependency, and future hospitalizations [1]. Further, these older adults are at risk for longer lengths of stay that have both physical (e.g., nosocomial infections, further functional decline) [2,3] and economical (e.g., increased hospital stays and resource utilization) [4] detriments. Comprehensive geriatric assessments (CGAs) use a multi-dimensional approach to examine older adults and have included models in which older adult inpatients are assessed by a mobile team [5]. Ontario has developed a similar model of care known as assess and restore (A&R) programs, which aim to prevent these adverse outcomes by providing short-term restorative care [6]. Humber’s Elderly Assess and Restore Team (HEART) is one such program utilizing physiotherapists, occupational therapists, rehabilitation assistants, and registered practical nurses to provide restorative care to hospitalized older adults at risk for functional decline. Enrolled participants are assessed within 48 h of hospital admission, given tailored therapy plans, and provided daily rehabilitative care. Upon discharge from the hospital, participants are connected to appropriate resources and contacted at home to ensure they have accessed these resources. Previous literature has described the program in further detail and has shown its success in reducing hospital resources and returning patients to their homes [7]. Currently, the HEART program serves approximately 14% of eligible participants due to limited resources. The limited program capacity and lack of predictive tools specific to A&R results in clinicians having to use their best judgement as to who to enroll. However, it is unknown if patients with certain characteristics derive greater benefit from the program compared to others while optimizing length of stay.

Research has typically focused on the role of individual factors or hospitalization processes in adverse outcomes in older adults. Some literature has suggested that functional decline in older adults is associated with characteristics such as dwelling [8], cohabitation [9], age [9], comorbidities [9], and falling [8,9]. Further literature has identified individual risk factors such as sex [10], comorbidities [10], marital status [11], and ambulation profile [11,12] as predictive of length of stay. When considering readmission to the hospital, marital status [11], ambulation profile [12], sex [13], and comorbidities [13] have been highlighted as predictors. Lastly, some of the literature has noted that discharge to home is less likely in those who live alone [11], use an ambulatory device [14], and have difficulty walking [14]. However, no research has examined these associations specifically in the context of an A&R program. The objective of this study was to determine whether certain patient characteristics are predictive of success (functional improvement, reduced length of stay, discharge to home, and decreased 30-day hospital readmissions) in the HEART program. By determining these characteristics, clinicians can identify patients most at risk for adverse outcomes and enroll those who would benefit most from the program, optimizing the use of resources and enhancing capacity.

## 2. Materials and Methods

### 2.1. Study Participants and Design

This study used a retrospective observational design to examine which predictors are associated with success in an A&R program. Participants were patients admitted to inpatient medicine at a community hospital in Toronto, Canada, between 4 September 2018 and 31 March 2020 who were eligible for the HEART program (aged ≥ 65 years, weight bearing, not a cerebrovascular accident, community-dwelling). To minimize selection bias, all patients eligible for the HEART program during this period (*n* = 6087) were considered. Participants missing covariate data were removed from analysis (*n* = 365). During this period, 547 patients participated in the HEART program. These participants were then matched to 547 eligible non-participants for a total sample of 1094 participants. This exceeded the sample size recommendation of 240 participants calculated by G*Power 3.1.9.7, which was computed with a medium effect size (f = 0.25), an alpha level of 0.05 (two-tailed), and power at 0.80. This power calculation was done for each outcome variable, and the largest sample size (i.e., 240 participants) was chosen as the sample size for the whole study. This study received ethical approval from Veritas Institutional Review Board, which waived the need for informed consent.

### 2.2. Predictor Variables

The study predictor variables were ambulatory device, ambulatory level, sex, age, comorbidities, condition, falls risk, place of dwelling, and marital status. As noted above, each of these variables have been highlighted as predictors of at least one measure of success (functional improvement, decreased length of stay, discharge to home, and decreased 30-day hospital readmissions) and were therefore chosen as potential predictors of success within the context of an assess and restore model. All predictor variables were determined through the examination of electronic health records. Predictor data were collected in the same manner for both the HEART and non-HEART participants. To prevent interrater variability, only one researcher (J.S.) analyzed health records. Participants were removed from analysis if they were missing covariate data.

Ambulatory device and ambulatory level were determined through physiotherapy and occupational therapy initial assessment notes. Ambulatory device was categorized as no device, cane, rollator walker, two-wheeled walker, “other” walker (high wheeled walker or standard walker), or “other” device (e.g., crutch). A participant’s ambulatory level was defined as independent, supervision, one-person assist (one-assist), or two-person assist (two-assist).

Hospital administrative databases were used to gather the patient’s sex, age, number of comorbidities, and their primary condition at admission. Their condition was defined as their Health Based Allocation Model Inpatient Group (HIG), a methodology used by the Ontario Ministry of Health to group similar conditions [15]. The seven most common HIGs were examined, with all other conditions grouped into another category called “other”.

Falls risk was determined using the Morse Fall Scale, a tool that has shown good specificity and sensitivity in hospitalized adults [16,17]. A nurse measured this at admission and defined patients as low-, medium-, or high-risk. Place of dwelling (home or retirement home) and marital status (single and lives alone, single and lives with family/caregiver, or married/living with partner) were determined from the notes of various clinicians (i.e., social worker, physiotherapist, physician).

### 2.3. Outcome Variables

The functional status of the participant was determined using the Barthel Index, a tool measuring functional independence that has shown high agreement amongst raters [18]. The Barthel Index score ranges from 0 to 100, with higher scores indicating greater independence [19]. The index has previously shown a minimal important change of 3.1 points at a 95% confidence interval in older adults [20]. A HEART physiotherapist or occupational therapist measured Barthel Index score at both hospital admission and discharge. Barthel Index was only measured in HEART participants. Difference in functional status was defined as the percent change in Barthel score from admission to discharge.

Hospital administrative databases were examined to determine patient length of stay, expected length of stay, discharge destination, and readmission to the hospital. Length of stay was defined as the total number of days spent in the hospital. The expected length of stay was defined as the number of days the participant was anticipated to stay in the hospital as calculated through HIG methodology [15]. Excessive length of stay (eLOS) was then determined as the difference between expected length of stay and the patient’s total length of stay in the hospital. Return to home was defined as a discharge to private residence/retirement home (i.e., not long term care, rehabilitative center, or another hospital unit). Readmission to the hospital was defined as admission to the hospital for the same condition within 30 days of discharge. This was gathered from hospital administrative databases that used coded data indicating whether readmission was for the same condition.

### 2.4. Statistical Analysis

We conducted propensity score matching to compare HEART participants to similar non-participants based on the aforementioned predictor variables. The MatchIt package in R, version 4.1.0 (The Comprehensive R Archive Network, http://cran.r-project.org, accessed on 12 January 2022), was used to conduct 1:1 nearest neighbor matching with a caliper of 0.2 [21]. We assessed covariate balance after matching using standard mean differences and a threshold of 0.1 [22]. All other statistical analyses were conducted using SPSS Version 28 (IBM Corp., Armonk, NY, USA). Standard descriptive statistics (means and standard deviations for continuous variables and percentiles for categorical variables) described participants. Functional change and eLOS were tested for normality using the Shapiro–Wilk test, with a *p*-value of < 0.05 suggesting that a nonparametric test is warranted. Differences in functional change across predictor levels in HEART participants were determined using one-way analyses of variance (ANOVAs) with Tukey post hoc tests. *p*-Values < 0.05 were considered statistically significant. Pearson correlation analyses were conducted to determine whether age and number of comorbidities were related to functional change. Since Barthel Index was only measured in HEART participants, differences in functional change between HEART and non-HEART participants based on predictor variables could not be determined. The Shapiro–Wilk test indicated a lack of normality for eLOS, so a series of Mann–Whitney tests were conducted to determine the differences in eLOS between HEART and non-HEART participants stratified by predictor variables. The associations between HEART participation and both readmission and return to home were determined using chi-square tests stratified by each predictor variable. In each test, *p*-values < 0.05 were considered statistically significant.

## 3. Results

Table 1 shows the descriptive characteristics of the 1094 participants included in the final analyses. Covariate balance was assumed, as all covariates had standardized mean differences of less than 0.1. Out of the 1094 included participants, the mean age of participants was ~84 years, and more than half (~61%) were female. There was an average change in Barthel score of 36.57% in HEART participants, with 99.2% of program participants maintaining or improving their functional capacity.

Figure 1 presents the differences in functional change based on predictor variables. Functional change was not significantly related to marital status, falls risk, dwelling, sex, age, or number of comorbidities. However, the condition of the patient affected functional change, *F*_7, 522_ = 2.28, *p* = 0.029, η_p_^2^ = 0.051, but Tukey post hoc testing revealed no significant differences between groups. Ambulatory status also had a significant effect on functional change, *F*_3, 524_ = 4.33, *p* = 0.005, η_p_^2^ = 0.024, with those who required a one-assist, having a higher functional change (M = 39.56, SD = 18.79) compared to those requiring supervision (M = 33.77, SD = 16.63, *p* = 0.003). There was also an effect of ambulatory device on functional change, *F*_4, 525_ = 7.37, *p* < 0.001, η_p_^2^ = 0.053, with those who did not use a device having a lower change (M = 26.18, SD = 12.94) compared to those using a rollator walker (M = 37.18, SD = 17.48, *p* = 0.006), two-wheeled walker (M = 37.76, SD = 18.51, *p* < 0.001), or “other” walker (M = 46.07, SD = 17.46, *p* < 0.001).

The results of the eLOS analyses are presented in Table 2. There was a significantly different eLOS between HEART participants and non-participants among those with an ambulatory level of one-assist (*p* = 0.002) or supervision (*p* = 0.001). This difference was greatest among those who were one-assist, with HEART participants having a median of 2.0 fewer days than non-participants. When stratified for condition, the association between HEART participation and eLOS were significant for those with pneumonia (*p* = 0.020), lower urinary tract infection (UTI) (*p* = 0.023), and “other” conditions (*p* < 0.001). The median difference (Mdn) in eLOS was greatest in those with a lower UTI (Mdn = 2.25). The association also existed in those with a high or low falls risk but not medium, with a Mdn of 1.80 days (*p* < 0.001) and 1.10 days (*p* = 0.039), respectively. There was a significantly different eLOS based on HEART participation in those with a rollator walker (Mdn = 2.20, *p* = 0.007), two-wheeled walker (Mdn = 1.90, *p* < 0.001), and “other” walker (Mdn = 3.80, *p* = 0.049). However, the association between eLOS and HEART participation did not hold for other ambulatory devices. The association held in those from home (Mdn = 1.30, *p* < 0.001) or a retirement home (Mdn = 1.00, *p* = 0.039). Both males and females had a significant association between HEART participation and eLOS (Mdn = 1.95, *p* < 0.001; Mdn = 0.90, *p* = 0.005, respectively).

Due to the low number of 30-day readmissions among HEART participants (*n* = 38), several categories had to be collapsed. Readmission to the hospital was not associated with HEART status stratified by any predictor variable (Table 3).

Few HEART participants did not return to community living (*n* = 56), so categories for some predictor variables had to be collapsed. The association between return to home and HEART participation held for stratified levels of all predictor variables except ambulatory status (Table 4). When stratified by ambulatory status, the association between HEART participation and return to home was not significant in those who were independent (*p* = 0.077) or were a two-assist (*p* = 0.806).

## 4. Discussion

We observed that participants who used a walker, required a one-person assist, were at a high risk of falling, and had either a lower UTI or pneumonia seemed to derive the greatest benefit from the program. Functional change was affected by condition, ambulatory device, and ambulatory level, with those using a walker and those who were a one-person assist having the greatest functional improvements. Differences in eLOS were greatest in those who had a lower UTI or pneumonia, lived at home, used a walker, required a one-person assist, and were a high falls risk. Readmission to the hospital within 30 days was not associated with any predictor variables. Return to home was associated with HEART participation in all predictor variables except for ambulatory status, where there was no association in those who were independent or required a two-person assist.

Our findings align with the literature regarding functional decline in hospitalized older adults. In hospital, mobility has been highlighted as protective against functional decline [3,23,24], which aligns with our findings that HEART participants largely maintained functional capacity. However, we found that using a walker was associated with greater functional improvement in the HEART program, which is contrary to one study, which discovered that those who used a walking device prior to hospitalization had a 2.81-times greater likelihood of functional decline compared to those without a device [25]. However, this difference could be due to participation in an A&R program, as those with a walker may have been functionally diminished at baseline and at risk for further decline without intervention. The authors could not find any literature that examined specific ambulatory devices or ambulatory status and their associations with functional decline. Thus, this study is likely the first to show that these factors may be associated with functional change in hospitalized older adults. The remaining findings in this study report differences in length of stay, readmission, and discharge destination between those who participated and those who did not participate in the HEART program based on predictor variables. Thus, these findings are difficult to compare to other literature examining predictor variables and adverse outcomes in older adults.

This study was not without limitations. Participants could not be perfectly matched to all characteristics due to the number of predictor variables used, so some bias may have occurred. Additionally, differences in eLOS based on HEART status could not be analyzed for statistical significance across predictor levels. We were limited to examining the eLOS results across predictors descriptively, so results should be interpreted with caution. For example, we found that males had a greater difference in eLOS compared to females, but it should be noted that both had decreased eLOS and would benefit from the program. Further, researchers were unable to determine whether there were differences in functional status between HEART and non-HEART participants due to the lack of Barthel information for non-participants. Future research should examine the differences in functional change between A&R participants and non-participants based on predictor variables. It is also possible that some hospital readmissions were missed, as data were limited to administrative databases at the study hospital and did not capture readmissions to other hospitals. Additionally, an association between readmission and HEART participation may not have been detected due to the low number of readmissions. The low number of readmissions also meant that categories of some predictor variables could not be studied (e.g., condition) or had to be collapsed for analysis, so further details about the predictors could not be gathered. Next, the retrospective nature of the data meant that researchers were limited to the information collected during hospitalization. This meant that certain characteristics (e.g., ethnicity) were missed and that other characteristics (e.g., dwelling) were reported with varying degrees of detail. This may have been especially true of ambulatory characteristics, as the participant’s ambulatory status or device may have differed based on the clinician assessing the patient and at which point they were seen. Finally, it is important to note that this program is geared towards a subset of frail older adults, so it is unknown if findings would apply to non-frail populations.

Our findings have implications for future healthcare practices. The results highlight which patients may benefit the most from an A&R program and give clinicians data-driven recommendations for enrollment. Our findings suggest that eLOS and functional improvement may be the most important outcomes to consider when enrolling participants in the program. Since readmission did not differ based on predictor variables, and discharge destination only differed based on ambulatory status, studied characteristics would likely not affect these outcomes (except ambulatory status and return to home). The authors suggest that predictors noting greater differences in functional status and eLOS should be considered when enrolling participants into A&R programs in Ontario. By enrolling the optimal patient, there may be a reduction in hospital costs and resource utilization, an increased program capacity to serve patients most in need of this service, an optimized functional status, and an improvement in patient and provider experience (i.e., quadruple aim). Other hospitals might consider implementing similar programs and using the predictors identified in this study as criteria when enrolling patients where there is limited capacity to services.

## 5. Conclusions

Little is known about predictors of success in the Ontario model of an A&R program. In hospitalized older adults taking part in the HEART program, several characteristics were related to functional decline, eLOS, and return to home. Greatest functional improvements were seen in those using a walker and requiring a one-person assist. Greatest decreases in eLOS were found in those using a walker, requiring a one-person assist, at a high falls risk, and admitted with either a lower urinary tract infection or pneumonia. This novel research identifies potential factors for consideration during enrollment into an A&R program, which had previously been unexamined.

## Figures and Tables

**Figure 1 geriatrics-07-00050-f001:**
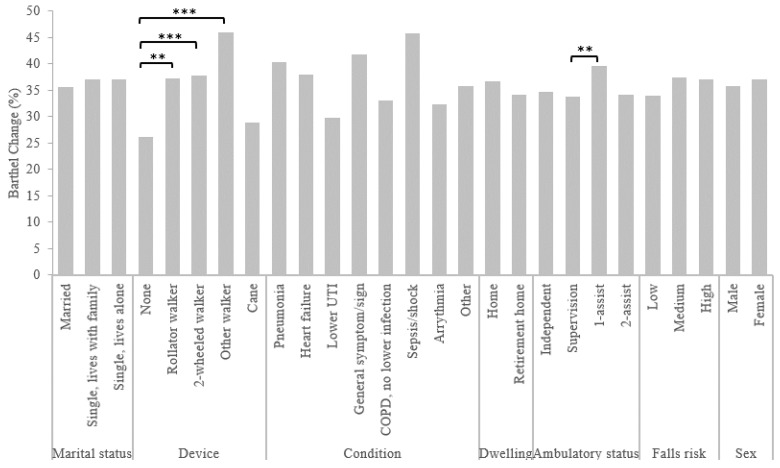
Functional change from admission to discharge based on chosen predictors in HEART participants. ** *p* < 0.01, *** *p* < 0.001. UTI, urinary tract infection; COPD, coronary obstructive pulmonary.

**Table 1 geriatrics-07-00050-t001:** Descriptive characteristics of study participants before and after propensity score matching.

	Unmatched HEART Eligible, Non-Participant (*n* = 5175)	Matched HEART Eligible, Non-Participant (*n* = 547)	Matched HEART Participants (*n* = 547)	Standardized Mean Differences
Female sex, *n* (%)	2636 (50.9)	337 (61.6)	332 (60.7)	0.02
Age (years), mean (SD)	81.1 (8.5)	83.9 (7.9)	83.8 (7.2)	−0.007
Marital status, *n* (%)				−0.03
Married/living with partner	2240 (43.3)	206 (37.7)	192 (35.1)	
Single, lives with family	1275 (24.6)	141 (25.8)	155 (28.3)	
Single, lives alone	1516 (29.3)	200 (36.6)	200 (36.6)	
Missing	144 (2.8)	0	0	
Pre-admission dwelling, *n* (%)				0.03
Home	4836 (93.4)	499 (91.2)	494 (90.3)	
Retirement home	339 (6.6)	48 (8.8)	53 (9.7)	
Conditions, *n* (%)				−0.008
Viral/unspecified pneumonia	311 (6.0)	56 (10.2)	57 (10.4)	
Heart failure without coronary angiogram	542 (10.5)	90 (16.5)	82 (15.0)	
Lower urinary tract infection	166 (3.2)	27 (4.9)	36 (6.6)	
General symptom/sign	180 (3.5)	29 (5.3)	26 (4.8)	
COPD without lower respiratory infection	171 (3.3)	17 (3.1)	24 (4.4)	
Other/unspecified sepsis/shock	144 (2.8)	12 (2.2)	14 (2.6)	
Arrhythmia without coronary angiogram	150 (2.9)	15 (2.7)	10 (1.8)	
Other	3511 (67.8)	301 (55.0)	298 (54.5)	
Ambulatory status, *n* (%)				−0.03
Independent	1261 (24.4)	81 (14.8)	53 (9.7)	
Supervision	1743 (33.7)	219 (40.0)	220 (40.2)	
1-person assist	547 (10.6)	183 (33.5)	257 (47.0)	
2-person assist	1383 (26.7)	64 (11.7)	17 (3.1)	
Missing	421 (4.7)	0	0	
Ambulatory device, *n* (%)				0.01
None	1774 (34.3)	112 (20.5)	50 (9.1)	
Rollator Walker	546 (10.6)	66 (12.1)	78 (14.3)	
2-Wheeled Walker	1738 (33.6)	246 (45.0)	372 (68.0)	
Other Walker	567 (11.0)	75 (13.7)	24 (4.4)	
Cane	291 (5.6)	42 (7.7)	23 (4.2)	
Other	75 (1.4)	6 (1.1)	0	
Missing	184 (3.6)	0	0	
Falls risk, *n* (%)				−0.005
Low	1389 (26.8)	106 (19.4)	101 (18.5)	
Medium	1367 (26.4)	152 (27.8)	164 (30.0)	
High	2416 (42.2)	289 (52.8)	282 (51.6)	
Missing	3 (0.1)	0	0	
Admission Barthel Score, mean (SD)	-	-	46.03 (15.56)	
Change in Barthel Score, mean (SD)	-	-	36.57 (18.22)	
Patients who maintained or improved Barthel score from admission to discharge, *n* (%)	-	-	543 (99.2)	

COPD, chronic obstructive pulmonary disorder; HEART, Humber’s Elderly Assess and Restore Team; SD, standard deviation.

**Table 2 geriatrics-07-00050-t002:** Excess length of stay in HEART and non-HEART participants based on predictor variables, median (IQR).

	Difference in Median eLOS	HEART Participants (*n* = 547)	HEART Eligible, Did Not Participate (*n* = 547)	*p*-Value
Marital Status				
Married/living with partner	1.40	0.00 (3.95)	1.40 (8.60)	**<0.001**
Single, lives with family	1.30	−0.20 (4.20)	1.10 (8.70)	0.079
Single, lives alone	1.00	0.70 (5.95)	1.70 (9.10)	**0.014**
Condition				
Viral/unspecified pneumonia	1.70	−0.40 (5.60)	1.30 (8.40)	**0.020**
Heart failure without coronary angiogram	−0.65	0.70 (5.73)	0.05 (8.80)	0.631
Lower urinary tract infection	2.25	−0.35 (2.45)	1.90 (5.90)	**0.023**
General symptom/sign	1.10	0.50 (5.10)	1.60 (14.10)	0.679
COPD without lower respiratory infection	0.60	0.00 (3.70)	0.60 (6.40)	0.138
Other/unspecified sepsis/shock	−1.30	1.10 (5.10)	−0.20 (32.80)	0.897
Arrhythmia without coronary angiogram	−0.40	−0.10 (4.10)	−0.50 (3.30)	0.934
Other	1.75	0.25 (5.03)	2.00 (9.50)	**<0.001**
Dwelling				
Home	1.30	0.00 (5.90)	1.30 (8.90)	**<0.001**
Retirement home	1.00	1.30 (6.20)	2.30 (9.70)	**0.039**
Ambulatory Device				
None	0.40	−1.10 (4.40)	−0.70 (4.50)	0.602
Rollator walker	2.20	−0.25 (5.28)	1.95 (8.40)	**0.007**
2-wheeled walker	1.90	0.40 (4.97)	2.30 (9.00)	**<0.001**
Other walker	3.80	1.90 (7.70)	5.70 (14.10)	**0.049**
Cane	−0.35	−0.70 (2.40)	−1.05 (4.60)	0.821
Other	-	-	−0.65 (3.60)	-
Ambulatory status				
Independent	−1.00	0.30 (5.35)	−0.70 (4.10)	0.548
Supervision	1.30	−0.20 (4.47)	1.10 (7.00)	**0.001**
1-assist	2.00	0.30 (4.70)	2.30 (9.70)	**0.002**
2-assist	5.75	3.40 (8.60)	9.15 (22.20)	0.120
Falls risk				
Low risk	1.10	−0.30 (4.40)	0.80 (7.90)	**0.039**
Medium risk	0.00	0.70 (5.05)	0.70 (5.10)	0.346
High risk	1.80	0.10 (4.80)	1.90 (9.10)	**<0.001**
Sex				
Male	1.95	−0.05 (5.08)	1.90 (9.60)	**<0.001**
Female	0.90	0.30 (4.65)	1.20 (8.40)	**0.005**

COPD, chronic obstructive pulmonary disorder; HEART, Humber’s Elderly Assess and Restore Team; SD, standard deviation.

**Table 3 geriatrics-07-00050-t003:** Readmission to the hospital in HEART and non-HEART participants based on predictor variables, *n* (%).

	HEART Participants (*n* = 38)	HEART Eligible, Did Not Participate (*n* = 37)	*p-*Value
Marital status			
Married/living with partner	12 (6.3)	15 (7.3)	0.683
Single, lives with family	11 (7.3)	8 (5.7)	0.618
Single, lives alone	15 (7.5)	14 (7.0)	0.847
Dwelling			
Home	34 (6.9)	32 (6.4)	0.766
Retirement home	4 (7.5)	5 (10.4)	0.613
Ambulatory device			
Rollator walker	5 (6.4)	2 (3.0)	0.347
2-wheeled walker	26 (7.0)	21 (8.5)	0.478
Other	7 (7.2)	14 (6.0)	0.668
Ambulatory status			
Independent	4 (7.5)	7 (8.6)	0.821
Supervision	17 (7.7)	14 (6.4)	0.585
1-assist	17 (6.7)	13 (7.1)	0.841
2-assist	0	3 (4.7)	0.363
Falls risk			
Low	9 (8.9)	5 (4.7)	0.230
Medium	11 (6.8)	13 (8.6)	0.536
High	18 (6.4)	19 (6.6)	0.926
Sex			
Male	15 (7.1)	19 (8.8)	0.520
Female	23 (6.8)	18 (5.4)	0.449

HEART, Humber’s Elderly Assess and Restore Team.

**Table 4 geriatrics-07-00050-t004:** HEART and non-HEART participants who did not return home based on predictor variables, *n* (%).

	HEART Participants (*n* = 56)	HEART Eligible, Did Not Participate (*n* = 133)	*p*-Value
Marital status			
Married/living with partner	17 (8.9)	51 (24.8)	**<0.001**
Single, lives with family	17 (11.0)	31 (22.0)	**0.010**
Single, lives alone	22 (11.0)	51 (25.5)	**<0.001**
Dwelling			
Home	49 (9.9)	118 (23.6)	**<0.001**
Retirement home	7 (13.2)	15 (31.3)	**0.028**
Ambulatory device			
Rollator walker	7 (9.0)	18 (27.3)	**0.004**
2-wheeled walker	36 (9.7)	61 (24.8)	**<0.001**
Other	13 (13.4)	54 (23.0)	**0.048**
Ambulatory status			
Independent	7 (13.2)	21 (25.9)	0.077
Supervision	24 (10.9)	57 (26.0)	**<0.001**
1-assist	22 (8.6)	42 (23.0)	**<0.001**
2-assist	3 (17.6)	13 (20.3)	0.806
Falls risk			
Low	7 (6.9)	24 (22.6)	**0.002**
Medium	24 (14.9)	39 (25.7)	**0.014**
High	25 (8.9)	70 (24.2)	**<0.001**
Sex			
Male	24 (11.4)	55 (25.6)	**<0.001**
Female	32 (9.5)	78 (23.5)	**<0.001**

HEART, Humber’s Elderly Assess and Restore Team.

## Data Availability

Not applicable.

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
