# Peer review of "Predictors of Functional Improvement, Length of Stay, and Discharge Destination in the Context of an Assess and Restore Program in Hospitalized Older Adults"

_geriatrics, 2022, doi:10.3390/geriatrics7030050_

Round 1

Reviewer 1 Report

The aim of this study was to determine which Humber’s Elderly Assess and Restore Team (HEART) participant characteristics are predictive of functional improvement, decreased length of stay, return to home, and decreased readmission to hospital.

Differences in functional status, excessive length of stay, discharge destination, and hospital readmissions were compared in 547 HEART patients and 547 matched eligible non-participants using ANOVAs, Mann-Whitney and chi-square tests.

The manuscript is very interesting and well written. I would like to suggest to the authors some changes in the statistical analysis to improve the obtained results.

As regard sample size estimation, how effect size was determined? For each outcome variable, the authors should specify the effect size (on the basis or relevant literature). Therefore, for each outcome there will be an estimated sample size, and the largest one could be considered the sample size for the whole study. Moreover, the model selected for the estimation and the number of tails should be specified for each estimation.

 Authors set a minimum power of 80%. Since the actual sample size is defectively larger than expected, authors could calculate the effective power they achieved din their study.

For continuous variables authors declared to use parametric statistics (mean±sd; ANOVA, pearson correlation), with the exception for eLOS, where authors stated they preferred using Mann –Whitney (which is non parametric) … Did they test the normality in distribution of their continuous variables (e.g. with Kolmogorov-Smirnof of Shapiro-Wilk test)?  If they did, they should declare it in the data analysis section. If not, I suggest to do it and apply the appropriate statistics according to the distribution of each variable.

Table 2: authors did not report the sample size for each group…is it still 547 v. 547?

Did the authors try to run any multivariable models?

In my opinion they could try as follow (including the results in just one table):

- linear regression on Barthel change including group (HEART/non-HEART) as independent variable and all the other variables as potential confounders

- linear regression on eLOS including group (HEART/non-HEART) as independent variable and all the other variables as potential confounders

- logistic regression on readmission including group (HEART/non-HEART) as independent variable and all the other variables as potential confounders

- logistic regression on home return including group (HEART/non-HEART) as independent variable and all the other variables as potential confounders

Author Response

Point 1: The aim of this study was to determine which Humber’s Elderly Assess and Restore Team (HEART) participant characteristics are predictive of functional improvement, decreased length of stay, return to home, and decreased readmission to hospital.

Differences in functional status, excessive length of stay, discharge destination, and hospital readmissions were compared in 547 HEART patients and 547 matched eligible non-participants using ANOVAs, Mann-Whitney and chi-square tests.

The manuscript is very interesting and well written. I would like to suggest to the authors some changes in the statistical analysis to improve the obtained results.

Response 1: Thank you for taking the time to review our manuscript and provide valuable feedback.

Point 2: As regard sample size estimation, how effect size was determined? For each outcome variable, the authors should specify the effect size (on the basis or relevant literature). Therefore, for each outcome there will be an estimated sample size, and the largest one could be considered the sample size for the whole study. Moreover, the model selected for the estimation and the number of tails should be specified for each estimation.

Response 2: Due to the unique nature of this program, the authors could not find literature that was highly relevant for determining the effect size for each outcome variable. Therefore, the authors elected to ensure an appropriate sample size using a standard medium effect size of 0.25 (Cohen’s F). This was done for each outcome variable. 240 was the largest sample size from these tests and therefore was chosen as the sample size for the whole study. The authors agree that the number of tails should be specified and included this in the revised manuscript (line 78) as well as explained that the 240 referred to the largest sample size per outcome (lines 78-80).

Point 3: Authors set a minimum power of 80%. Since the actual sample size is defectively larger than expected, authors could calculate the effective power they achieved din their study.

Response 3: The authors chose not to calculate the effective power achieved in the study to prevent the appearance of data mining after the sample size had been set.

Point 4: For continuous variables authors declared to use parametric statistics (mean±sd; ANOVA, pearson correlation), with the exception for eLOS, where authors stated they preferred using Mann –Whitney (which is non parametric) … Did they test the normality in distribution of their continuous variables (e.g. with Kolmogorov-Smirnof of Shapiro-Wilk test)?  If they did, they should declare it in the data analysis section. If not, I suggest to do it and apply the appropriate statistics according to the distribution of each variable.

Response 4: The authors tested the normality in the distribution of our continuous variables using the Shapiro Wilk test. We have added a statement indicating this in the data analysis section in the revised manuscript:

Lines 139-141: Functional change and eLOS were tested for normality using the Shapiro Wilk test, with a P value of < .05 suggesting that a nonparametric test is warranted.

Lines 147-150: The Shapiro Wilk test indicated a lack of normality for eLOS, so a series of Mann-Whitney tests were conducted to determine the differences in eLOS between HEART and non-HEART participants stratified by predictor variables.

Point 5: Table 2: authors did not report the sample size for each group…is it still 547 v. 547?

Response 5: Thank you for your comment. Yes, it is still 547 v 547. We have added this information to Table 2 in the revised manuscript.

Point 6: Did the authors try to run any multivariable models?

In my opinion they could try as follow (including the results in just one table):

- linear regression on Barthel change including group (HEART/non-HEART) as independent variable and all the other variables as potential confounders

- linear regression on eLOS including group (HEART/non-HEART) as independent variable and all the other variables as potential confounders

- logistic regression on readmission including group (HEART/non-HEART) as independent variable and all the other variables as potential confounders

- logistic regression on home return including group (HEART/non-HEART) as independent variable and all the other variables as potential confounders

Response 6: Thank you for your comment. The authors decided to run ANOVA, Mann-Whitney and Chi-square tests instead of linear and logistic regressions because the findings are more comprehensive for the readers. The authors also chose to conduct a propensity score match based on the other variables instead of running a model with them as confounders. The authors found a balance in covariates across the groups (standard differences <0.1), as noted in line 157, and decided not to further adjust for confounders. This was in accordance with findings from Nyugen et al. (2017, https://doi.org/10.1016/j.archger.2021.104609), who found that adjustment for confounders provided only a negligible benefit when there were standard differences less than 0.1.    

Reviewer 2 Report

Thank you for your contribution. I have read it with great interest. The objective of this study was to determine whether certain patient characteristics are predictive of success (functional improvement, reduced length of stay, discharge to home, and decreased 30-day hospital readmissions) in the HEART program. That will be helpful if we can find what characteristics can predict and improve the functions of older adults.

Introduction

On P.2 Line 58-59, as authors stated: “However, no research has examined these associations specifically in the context of an A&R program.” It shows that it is an important issue for older care. Could the author provide some literatures about how patient characteristics affect (functional improvement, reduced length of stay, discharge to home, and decreased 30-day hospital readmissions) in other program?

Materials and Methods

The authors should provide more information why they use these predictor variables and outcome variables.

Results

  1. On P.4 Line 149-150, the statement: “there was an average change in Barthel score of 36.57% in HEART participants, with 2% of program participants maintaining or improving their functional capacity.” Is this result (99.2%) from Table 1? Would authors can explain more about this?
  2. According these results, I’m not sure is it answer the question: “Predictors of Functional Improvement, Length of Stay, and Discharge Destination in the Context of an Assess and Restore Program in Hospitalized Older Adults.” In other word, I still don’t know what’s different between two group (547 patients participated in the HEART program and 547 eligible non-participants, for a total sample of 1094 participants) on functional improvement, length of stay, and discharge destination. I do think this is the key contribution of this paper—tell reader more about what’s different between participate A&R program and not participate A&R program.

Conclusion:

I do think that the paper has potential but a lot of more work needs to be done in order to strengthen the arguments being made.

Author Response

Point 1: Thank you for your contribution. I have read it with great interest. The objective of this study was to determine whether certain patient characteristics are predictive of success (functional improvement, reduced length of stay, discharge to home, and decreased 30-day hospital readmissions) in the HEART program. That will be helpful if we can find what characteristics can predict and improve the functions of older adults.

Response 1: Thank you for taking the time to review our article and provide valuable feedback.

Point 2: Introduction

On P.2 Line 58-59, as authors stated: “However, no research has examined these associations specifically in the context of an A&R program.” It shows that it is an important issue for older care. Could the author provide some literatures about how patient characteristics affect (functional improvement, reduced length of stay, discharge to home, and decreased 30-day hospital readmissions) in other program?

Response 2: In lines 50-58, the authors note studies in which patient characteristics have been shown to affect these outcomes. Some of these studies were conducted in the context of a program. In the discussion, the authors also note some mobility programs and how these have affected outcomes.

Point 3: Materials and Methods

The authors should provide more information why they use these predictor variables and outcome variables.

Response 3: Thank you for your comment. The authors have added the following statement to describe why these variables were chosen:

Lines 84-88: As noted above, each of these variables have been highlighted as predictors of at least one measure of success (functional improvement, decreased length of stay, discharge to home, and decreased 30-day hospital readmissions) and were therefore chosen as potential predictors of success within the context of an assess and restore model.

Point 4: Results

  1. On P.4 Line 149-150, the statement: “there was an average change in Barthel score of 36.57% in HEART participants, with 2% of program participants maintaining or improving their functional capacity.” Is this result (99.2%) from Table 1? Would authors can explain more about this?

Response 4: This result was not presented in the table and was only added as descriptive information. The 99.2% refers to the percentage of HEART patients whose Barthel score was the same or higher upon discharge as it was at admission. We added a row to Table 1 to indicate this.

Point 5:

  1. According these results, I’m not sure is it answer the question: “Predictors of Functional Improvement, Length of Stay, and Discharge Destination in the Context of an Assess and Restore Program in Hospitalized Older Adults.” In other word, I still don’t know what’s different between two group (547 patients participated in the HEART program and 547 eligible non-participants, for a total sample of 1094 participants) on functional improvement, length of stay, and discharge destination. I do think this is the key contribution of this paper—tell reader more about what’s different between participate A&R program and not participate A&R program.

Response 5: The authors have previously published an article whose aims were to determine whether those in HEART differed from similar non-participants and cited this in the current article (Edelstein and Scandiffio, 2022, https://doi.org/10.1016/j.archger.2021.104609). This was done intentionally to ensure a lack of overlap between the manuscripts. The aim of this paper was to determine which variables are predictive of optimal success within this program (i.e. which participants are best suited to the program) and not whether the program itself is more likely to improve clinical outcomes.

Point 6: Conclusion:

I do think that the paper has potential but a lot of more work needs to be done in order to strengthen the arguments being made.

Response 6: Thank you for your review.

Reviewer 3 Report

Dear Authors,

 I believe that the manuscript has high potential for publication. Please, see my comments and suggestions below.

Introduction has an adequate background and the importance of the study. Materials and Methods are clear. Results are well presented. In Discussion, I suggest delete the first phrase “To the knowledge of the authors, this is the first study to examine which characteristics predict success in an program in hospitalized older adults” (Line 212-213), because you repeat this information in Line 234-235: “Thus, this study is likely the first to show that these factors may be associated with functional change in hospitalized older adults.” The discussion already shows the need for further research. Then, you do not need to repeat it again in conclusion. I recommend presenting your main results in conclusion and finishing it by highlighting the novelty of your study and its importance to the health practice.

Author Response

 I believe that the manuscript has high potential for publication. Please, see my comments and suggestions below.

Response 1: The authors would like to thank you for taking the time to view our manuscript and provide your valuable comments and suggestions.

Point 2: Introduction has an adequate background and the importance of the study. Materials and Methods are clear. Results are well presented. In Discussion, I suggest delete the first phrase “To the knowledge of the authors, this is the first study to examine which characteristics predict success in an program in hospitalized older adults” (Line 212-213), because you repeat this information in Line 234-235: “Thus, this study is likely the first to show that these factors may be associated with functional change in hospitalized older adults.” The discussion already shows the need for further research. Then, you do not need to repeat it again in conclusion. I recommend presenting your main results in conclusion and finishing it by highlighting the novelty of your study and its importance to the health practice.

Response 2: Thank you for your comment. We have removed this sentence (line 221-222). We have also removed the sentence noting the need for further research in the conclusion and added our main findings in addition to the novelty of our findings.

Round 2

Reviewer 1 Report

I thank the authors for having considered my suggestions and having answered to all my points. In my opinion the article has improved and it is now suitable to be published in Geriatrics.

Author Response

Thank you for reviewing our article and providing your valuable feedback.